# Simulating Urban Expansion Based on Ecological Security Pattern—A Case Study of Hangzhou, China

**DOI:** 10.3390/ijerph19010301

**Published:** 2021-12-28

**Authors:** Xiaochang Yang, Sinan Li, Congmou Zhu, Baiyu Dong, Hongwei Xu

**Affiliations:** Institute of Applied Remote Sensing and Information Technology, College of Environmental and Resource Sciences, Zhejiang University, Hangzhou 310058, China; 21914154@zju.edu.cn (X.Y.); lisinan@zju.edu.cn (S.L.); congmouzhu1993@zju.edu.cn (C.Z.); 11914074@zju.edu.cn (B.D.)

**Keywords:** urban expansion, ecological security pattern, scenario simulation, CA-Markov-FLUS model, Hangzhou

## Abstract

Disordered urban expansion has encroached on a large amount of ecological land, resulting in the steady degradation of urban ecology, which has an adverse effect on the sustainable development of the region. An ecological security pattern can effectively control urban expansion, and it is of great significance to balance urban development and ecological protection. In order to analyze the impact of ecological security patterns on urban expansion, Hangzhou was taken as an example, the CA-Markov model and FLUS model were used to simulate the urban expansion pattern in 2030 under the natural development scenario and the ecological security scenario. The results showed that (1) the ecological source area in the study area is 630.90 km^2^ and was mainly distributed in the western mountainous area. There are 14 ecological corridors, primarily composed of valleys and rivers. Ecological nodes are mainly distributed on the north and south sides of the main urban area. (2) From 2000 to 2018, the annual increase index (AI) of construction land decreased in the northeast and southeast directions but increased in the northwest and southwest directions, and in the northeast direction the value was always the highest. Except for the southwest direction, the average annual growth rate (AGR) of construction land in the other directions decreased. At a distance from the city center of 30 km, AI was relatively higher and was increasing, while AGR was declining. At a distance of 30–45 km, both AI and AGR were increasing, indicating that the focus of construction land was moving outwards. (3) From 2018 to 2030, under both natural development scenario and ecological security scenario, construction land would keep expanding, but the construction land area, proportion, AI, and AGR of the latter would both be smaller than the former, indicating that the ecological security pattern can effectively curb urban expansion. Because of a large amount area of ecological sources, the expansion of construction land in the southwest direction would be constrained, especially under the ecological security scenario. The methods and results of this study can provide theoretical and application references for urban planning and green development in metropolises.

## 1. Introduction

Urban expansion is often manifested as construction land sprawl in unbuilt areas. Under the influence of the long-term policies of prioritizing economic development, a large amount of non-construction land has been converted into construction land, especially in urban areas. With the rapid expansion of construction land, a large area of ecological lands, such as agricultural land, vegetation, and water, has shrunk drastically [1]. In recent years, more and more scholars have noticed the negative ecological impact caused by urban expansion [2]: the expansion of built-up areas has led to the heat island effect [3,4], and there also exists a long-run equilibrium relationship between environmental pollution and urban expansion [5]. The Chinese government has realized the importance of sustainable development and has begun to advocate building an ecologically civilized society, which is a composite system of “society-economy-nature”, aiming to achieve harmonious co-existence between man and nature. Moreover, it has issued a series of strict policies to restrict the conversion of other lands to construction land (adding new construction land will require a package of complicated application procedures).

The ecological security pattern aims to identify important ecological regions and to ensure their connectivity, which is of great significance to the protection of regional ecological security [6,7]. With the acceleration of urbanization, urban expansion has changed the structure and process of ecosystems, resulting in a substantial reduction in ecosystem service functions and seriously threatening the sustainable development of cities and the quality of living in urban areas [8]. An ecological security pattern helps to identify important ecological space of the city and improves the efficiency of ecological protection. In the specific era of green mountains and clear water equal to mountains of gold and silver, building the urban ecological security pattern is positive to promoting high-quality urban development.

Currently, urban land use change simulation has been widely applied to spatial decision-making [9]. The current focus of land use change simulation research can be roughly divided into three categories. First, the study of simulation methods. Common models for land use change simulation include the CA-Markov model [10], FLUS model [11], CLUE-S model [12,13], PLUS model [14], etc. The main objective is to predict land demands, calculate the spatial suitability distribution probability of each unit, and then spatially distribute land demands according to spatial distribution probability. Furthermore, deep learning algorithms have also been applied to urban change simulations, such as the coupled bargaining model and modified ant colony optimization (ACO) algorithm [15]. The CA model integrated with deep learning (DL) techniques includes a convolutional neural network, a recurrent neural network and a random forest [16], and metaheuristic processes such as particle swarm optimization (PSO), generalized simulated annealing (GSA) and genetic algorithms (GAs) [17]. Second, adjusting and improving the simulation process, such as the simulation of land demands [18] and the simulation scale [19], to improve simulation accuracy. The third is to apply it to actual decision-making according to the results of land use simulation. Generally, multi-scenario simulations are carried out by changing the restrictions [20,21,22,23,24] to analyze the possible value of a specific target in different scenarios to support the spatial decision-making process. In addition, some scholars began to pay attention to the changes in specific land types in the city, such as built-up land density [25] and residential land growth [26].

The ecological security pattern is often regarded as a spatial restriction in the simulation process of land use change simulation [27,28]. However, current research generally pays attention to the impact of land use change on regional ecology in the long-term land use change process [29], and under the restriction of the ecological security pattern, evaluating whether the future regional ecological quality is better than the current and other scenarios. Few studies have focused on the possible impact of ecological security patterns on land use changes. Urban development is a sustaining process, and in the context of advocating the construction of ecological civilization, it is of great significance to balance urban development and ecological protection.

In view of the above considerations, this study took Hangzhou, a region of rapid economic development, as an example, and based it on ecological security patterns to simulate the urban expansion pattern under natural development and ecological security scenarios. Specifically, this study included three main parts: (1) constructing the ecological security pattern for the rapid economic development region; (2) identifying spatiotemporal variations in construction land expansion from 2000 to 2018; and (3) scenarios simulating construction land expansion based on ecological security patterns. This study provides a reference for the development of ecologically civilized cities.

## 2. Materials and Methods

### 2.1. Study Area

This study took the downtown of Hangzhou (Figure 1) as the study area, located in the south of the Yangtze River Delta in northeastern Hangzhou, Zhejiang Province, with a total area of 3349 km^2^. In May 2021, the highest government of China issued *the Opinions of the Central Committee of the Communist Party of China and the State Council on Supporting Zhejiang’s High-quality Development and Construction of a Common Prosperity Demonstration Zone* and *Zhejiang High-quality Development and construction of a Common Prosperity Demonstration Zone Implementation Plan (2021–2025)* to support Zhejiang Province in building a demonstration zone for common prosperity. Hangzhou, as the capital of Zhejiang Province, is the center of economic development in the province, and the downtown area of Hangzhou is the core economic development area of Hangzhou. As of 2018, the regional GDP was 10260.67 billion, accounting for more than 70% of the GDP of the city (data source: Hangzhou Bureau of Statistics). It also has the highest population density, which brings together more than 70% of the resident population of Hangzhou. Due to the demands of economic development and the sustaining influx of the floating population, there is a greater demand for land. The conflict between land supply and demand is prominent, and there is a risk of various ecological land being occupied for construction. Throughout the whole area of Hangzhou, the downtown area has less ecological land and fewer large-scale habitat patches, mainly distributed on the west side of the study area. It is necessary to think about how to keep the current ecological space and maximize the efficiency of ecological protection in the case of tight land.

### 2.2. Data Sources

In this study, land use data included three phases in 2000, 2010, and 2018, with a spatial resolution of 30 m, which came from the multi-period land use/land cover remote sensing monitoring database (CNLUCC) of the Chinese Academy of Sciences. DEM (digital elevation model) data came from the geospatial data cloud platform (http://www.gscloud.cn/ (15 November 2021)) with a spatial resolution of 30 m. Slope and topographic relation data were calculated from DEM data with the Slope tool and the Focal Statistics tool in ArcGIS 10.7. River, railway, highway, and residential data were acquired from the Natural Resources Department. Distances to rivers, railways, roads, and residential areas were calculated by the Euclidean Distance tool in ArcGIS10.7. Night light data came from DMSP/OLS and NPP/VIIRS data, and the two kinds of data were fitted. Population density data came from the WorldPop Asia dataset (https://www.worldpop.org/geodata (7 August 2021)), with a spatial resolution of 1000 m, and were resampled to 30 m with nearest neighbor interpolation method, which would retain original image information.

### 2.3. Construction of Ecological Security Pattern

#### 2.3.1. Identifying Ecological Sources and Nodes

The morphological spatial pattern analysis (MSPA) method can be used for characterizing binary patterns with an emphasis on connections between their parts [30]. Forest ecosystems and high-coverage grassland ecosystems have higher ecological service value and ecological stability. Based on the land use data in 2018, this study constructs the ecological security pattern of the study area, taking woodland and high-coverage grassland as the ecological prospect and other land types as the ecological background, and used the 8-neighbor algorithm to divide the ecological prospect into 7 non-overlapping landscape types, such as core and bridge. The landscape composition calculated via the MSPA method has an obvious scale effect, as the unit scale changes, the area of cores would change accordingly, which means that the edge width greatly impacts the area and shape of patches [31]. Given that the study area was small, the edge width was set to 1, corresponding to a 30 m actual distance. The cores greater than 100 hm^2^ were selected, and the values of the probability index of connectivity (dPC) were calculated to determine ecological sources. dPC measured the ability of each patch to maintain the overall connectivity of the landscape. The greater the dPC value is, the stronger the ability to maintain the overall connectivity of the landscape [32]. The calculation formulas are as follows:(1)$$\mathrm{PC}=\frac{\sum_{i=1}^{n}\sum_{j=1}^{n}p_{ij}\times a_{i}\times a_{j}}{A_{L}^{2}}$$
(2)$$\mathrm{dPC}=\frac{PC-PC_{remove,k}}{PC}$$

In the above formulas, n is the number of patches, $a_{i}$ and $a_{j}$ represent the areas of patches i and j, respectively, $p_{ij}$ represents the maximum connection probability in all paths between patches i and j, $A_{L}$ represents the total area of all patches, $PC_{remove,k}$ represents the landscape connectivity index after patch k is removed from the study area.

Conefor2.6 was used to calculate the dPC value of the cores with an area greater than 100 hm^2^, and the distance threshold was set as 1500 m and the connectivity probability as 0.5. Those cores with dPC > 1 were selected as ecological sources. In addition, water greater than 100 hm^2^ (except rivers) was added as the ecological source. The other cores were used as ecological nodes.

#### 2.3.2. Constructing Ecological Corridors

The minimum cumulative resistance (MCR) refers to the work done to overcome resistance from the “source” to different landscapes, reflecting the least-cost path from the target patch to the nearest source patch [33]. This path is the best path for the migration and spread of biological species. The calculation formula is as follows:(3)$$\mathrm{MCR}=f_{min}\sum_{j=n}^{i=m}\left(D_{ij}\times R_{i}\right)$$

In the formula, MCR is the minimum cumulative resistance value, $D_{ij}$ represents the spatial distance of the species from source j to landscape i, and $R_{i}$ represents the resistance coefficient of landscape unit i to species movement/migration.

Different land use/cover types have different resistances to species migration or landscape flows, and topographic relief also affects the distribution of landscape types [34], therefore, a minimum cumulative resistance surface was built from these two aspects, to calculate the minimum resistance paths. The gravity model can be used to quantitatively evaluate the interaction intensity between sources and target patches. When the interaction was greater, the material flow resistance between the source and the target patch was smaller [13]. Therefore, it can be used to select important corridors from generated ecological corridors to build ecological networks. The generated ecological corridor is lines, given that most of the ecological corridors were rivers or roads surrounded by protective forests, and the width of urban protective forests is generally 80–100 m, so a 100 m buffer zone was established based on the selected ecological corridors as real ecological corridors. In addition, rivers with a width of more than 30 m and a length of more than 5000 m in the study area were also regarded as ecological corridors. Ecological resistance values and weights were set by referencing related studies [35,36] and combining regional situations (Table 1). The calculation formula of the gravity model is as follows:(4)$$G_{ij}\frac{N_{i}N_{j}}{D_{ij}^{2}}=\frac{\left[\frac{1}{P_{i}}\times \ln\left(S_{i}\right)\right]\;\left[\frac{1}{P_{j}}\times \ln\left(S_{j}\right)\right]}{{\left(\frac{L_{ij}}{L_{max}}\right)}^{2}}=\frac{L_{max}^{2}\ln\left(S_{i}\right)\ln\left(S_{i}\right)}{L_{ij}^{2}P_{i}P_{j}}$$

In the formula, $G_{ij}$ is the interaction force between patches i and j, $N_{i}$ and $N_{j}$ are the weight of the two patches, respectively, $D_{ij}$ represents the normalized value of the potential corridor resistance between patches i and j, $P_{i}$ represents the resistance of patch i, $S_{i}$ represents the area of patch i, $L_{ij}$ represents the cumulative resistance value of the corridor between patches i and j, $L_{max}$ is the maximum cumulative resistance of all corridors.

#### 2.3.3. Land Use Change Simulation

Based on the system dynamics and cellular automata models, the FLUS model is established by integrating the artificial neural network algorithm (ANN) and the adaptive inertial competition mechanism based on roulette selection [11]. The model was composed of ANN-based probability-of-occurrence estimation and self-adaptive inertia and competition mechanism (CA) and had a relatively high prediction accuracy [14]. It has been widely used in studies of urban expansion simulation and practices of land use pattern optimization. The core idea of the FLUS model in land use simulation is to fit the land use data and driving factors based on the ANN to preliminarily calculate the spatial adaptive probability of each land class, and then, based on the adaptive probability, the adaptive inertia coefficient, neighborhood weight, conversion cost, and restricted area are added to calculate the joint distribution probability of different land classes on each unit. The FLUS model constructed an adaptive inertial competition mechanism based on the roulette selection algorithm. In the roulette algorithm, the probability of each land type being selected is directly proportional to the adaptability to adapt to the uncertainty of land use type transformation. The simulation result is finally generated via spatially optimal allocation [37].

Urban land use changes are significantly affected by natural and socioeconomic development. This study referred to the existing research [38] and selected 9 driving factors (Figure 2) from both natural and socio-economic aspects to calculate the probability of occurrence of different land types on each unit. Using the CA-Markov model to predict land demands, the forecast result is only affected by the previous stage [39]. The neighborhood parameters (Table 2) were set referring to the research of another scholar [40]. This study proposed two different future scenarios: a natural development scenario and an ecological security scenario. Under the natural development scenario, the conversion between some land types is prohibited (Table 3a) but without restricted areas. Under the ecological security scenario, in addition to limiting the mutual transformation of some land types (Table 3b), the constructed ecological security pattern was set as the restricted area. Compared with the natural development scenario, the ecological security scenario prohibited the conversion of woodland to arable land and construction land, and transformation from grassland to arable land was also prohibited (Table 2), where 0 indicated that conversion was not allowed, while 1 indicated that conversion was permitted.

### 2.4. Analysis of Urban Construction Land Expansion

Based on the existing land use data and simulation results, calculating the construction land center of gravity in 2018 as an urban center, the study area was divided into four quadrants and 12 multi-buffers at intervals of 5 km. The proportion of construction land, the annual increase index (AI), and the average annual growth rate (AGR) were calculated [3], to analyze the expansion characteristics of urban construction land in each quadrant and buffer zone quantitatively. The proportion of construction land is the proportion of construction land in the total area of regional land. AI can be used to compare the expansion intensity of construction land in different periods in the same region, and AGR reflects the expansion rate of construction land in different regions in the same period.

The calculation formulas are as follows:(5)$$\mathrm{AI}=\frac{A_{end}-A_{start}}{d}$$
(6)$$\mathrm{AGR}=100\%\times \left[{\left(\frac{A_{end}}{A_{start}}\right)}^{\frac{1}{d}}-1\right]$$

In the above formulas, AI represents the annual increase index, AGR represents the average annual growth rate, $A_{start}$ and $A_{end}$ are the construction land areas at the beginning and the end of the research unit, respectively, and d is the time interval.

## 3. Results

### 3.1. Ecological Security Pattern

Ecological sources were identified by the MSPA method and dPC value; given the actual situation, large water patches were included in the ecological sources. There are 21 large ecological patches with an area of 630.90 km^2^. Ecological sources were comprised of different land use types, and woodland was the majority, which accounted for 93.84% of the total area of the ecological source, while grassland accounted for less than 1%. In terms of spatial distribution (Figure 3), large ecological sources were distributed along the west side of the study area and extended to the areas on both sides of the eastern peninsula. This region was the area with a concentrated and contiguous distribution of woodland in the downtown of Hangzhou. In the east, the living and production spaces mainly consisted of construction land and arable land. It also contained scattered and relatively small forestland and grassland and other ecological lands. The ecological function decreased significantly from west to the east and from south to north. In ecological sources, the woodland was large and concentrated, while the grassland was scattered, which played a complementary and transitional role. A few ecological source patches were scattered in the east to supplement the ecological function. In terms of the spatial distribution of the importance of patches, those patches with the highest dPC value were distributed in the northwest corner of the study area. The second is the relatively large ecological patch located in the west and middle of the main urban area. Generally, the patches with higher ecological importance tend to be larger and closer to surrounding ecological patches.

The MCR model was used to build ecological corridors, and the resistance of each unit affected the final distance and direction of the corridors. In terms of spatial distribution, the area with the highest resistance value was the large and contiguous construction land in the middle of the study area, which was a great obstacle to species migration and energy flows. The area with the lowest resistance value was in the western part of the study area with a concentrated distribution of ecological sources. Based on the resistance surface, ecological corridors were constructed, and then the important corridors were selected via the gravity model. Because rivers are natural ecological corridors that play an important role in regional connectivity [41], several rivers, including the Qiantang River, were selected as ecological corridors to improve the overall connectivity of the pattern. A total of 14 ecological corridors were finally determined. These ecological corridors were mainly distributed in the western part of the study area, which played an important role in connecting various ecological source patches. The ecological exchange between the eastern and western parts of the study area was realized through the Qiantang River to a large extent.

Ecological nodes were key ecological strategic points used to bridge the links between adjacent ecological sources. The tangent points of the equal resistance lines between adjacent sources are often regarded as ecological nodes [16], and they can also be directly identified according to experience. To ensure the overall connectivity of the ecological network in the study area, a total of 7 ecological patches with relatively large areas distributed in the middle of the study area were selected as nodes to serve as “stepping stones” between different ecological source patches to help achieve the exchange of species and biological flows in the study area.

### 3.2. Urban Construction Land Expansion from 2000 to 2018

Comparing the data of urban construction land (Figure 4) in three phases (2000, 2010, 2018), urban construction land in Hangzhou expanded significantly from 2000 to 2018. From 2000 to 2010, the proportion of construction land in the study area increased from 12.29% to 19.71%, which was approximately 2/3 of the original construction land. By 2018, the area of construction land accounted for 25.86% of the whole area, more than double that of 2000. In terms of AI, the AI from 2000 to 2010 was 24.86, and the AI from 2010 to 2018 was 25.76, which showed that the urban expansion intensity of the study area increased from 2010 to 2018. Comparing the AGR of the two stages, it was found that the AGR in 2010–2018 was slightly less than that in 2000–2010, indicating that the urban expansion rate was slowing down.

In terms of the expansion direction (Figure 5a), in 2000 the construction land area in the northwest direction accounted for the highest proportion in the study area. After 2010, the proportion of construction land in the southeast direction was the highest. The old downtown of Hangzhou was in the northwest direction, while the government was committed to building a high-tech zone in the southeast direction. It was reasonable for the construction land to expand rapidly in the southeast direction. In 2018, the proportion of construction land in each direction was between 24% and 29%, the proportion of construction land in the northwest and southwest directions was slightly lower, and the urbanization level in each direction was relatively balanced.

In terms of AI (Figure 5b), in the northeast direction, the AI increased first and then decreased, but the value was always the highest. In the northwest direction, AI kept increasing, indicating that the expansion intensity of construction land was increasing. In the southwest direction, AI increased more obviously than that in the northwest direction. In the southeast direction, AI decreased significantly compared with the other directions, the expansion intensity of construction land also decreased significantly in this direction. In terms of AGR (Figure 5c), from 2000 to 2010 the AGR in the northeast direction was the highest, followed by the southeast direction. From 2010 to 2018, the AGR in the southwest direction increased slightly compared with the previous stage, but the value in the other three directions decreased, among which the AGR in the northeast direction decreased most significantly.

In 2000, the area of construction land (Figure 6a) within 5 km accounted for 40.21%. The proportion of construction land gradually decreased from the center to the outside. In 2010, the proportion of construction land was more than 50% within 0–10 km, and the area of construction land within 10–15 km almost doubled. In 2018, within 5 km, the proportion of construction land was 71.17%, and construction land was absolutely dominant. In the region within 50 km, the construction land in each range expanded to varying degrees. In terms of AI (Figure 6b), areas with high AI were concentrated within 30 km. From 2010 to 2018, the AI of other ranges except 10-15 km was generally higher than that from 2000 to 2010, indicating that the expansion intensity of construction land was increasing. AI in the range of 10–15 km decreased from 7.78 in the previous stage to 4.13. According to the land use data, there was a large area of woodland in this range, which hindered the expansion of construction land. In terms of AGR (Figure 6c), in 2000–2010, the values of construction land in the range of 10–20 km and 35–50 km were between 5% and 7%. From 2010 to 2018, the AGR of construction land within 5 km increased, but at 5–30 km, the AGR decreased. Overall, the annual growth rate of construction land within 30 km slowed down. Within the range of 30–45 km, the AGR of construction land increased, indicating that the expansion rate of construction land became faster in the area far from the center. In the area near the suburbs, construction land does not obviously change.

### 3.3. Urban Expansion Simulation Based on Ecological Security Patterns

#### 3.3.1. Validation

This study divided the region into six land use types: arable land, woodland, grassland, water, construction land, and unused land. On the basis of the land use data in 2010, according to the characteristics of area changes of various land types from 2000 to 2010, the actual land demands were spatially allocated via the FLUS model to acquire simulation results in 2018. The kappa coefficient was used to validate the simulation accuracy. Comparing simulation results with the actual land use data in 2018, 10% units were selected, and the kappa coefficient was calculated. The kappa coefficient was 0.841, and the overall classification accuracy was 0.890. Specifically, except grassland was slightly lower (0.784), the classification accuracies of other types were rational, furthermore, the accuracy of grassland, water area, and unused land was higher than 0.9. With high simulation accuracy, based on land use data in 2018, future land use simulations were acquired under different scenarios.

#### 3.3.2. Urban Expansion Simulation under Different Scenarios

There were no restrictions under the natural development scenario (Figure 7a) except that the conversion of some land types to water areas or unused land. In 2030, the area of construction land is expected to reach 1185.81 km^2^, accounting for 32.35% of the whole area, a net increase of 319.63 km^2^ over 2018. From 2018 to 2030, the AI is 26.55, slightly higher than that in 2010–2018 (Table 4), and would still maintain a high expansion intensity. The AGR is 2.64%, and the value gradually decreased in different time periods from 2000 to 2030.

Under the ecological security scenario (Figure 7b), in addition to restrictions on land conversion, the ecological security pattern was set as a restricted area to prohibit land transfer. In 2030 (Table 5), the area of construction land is expected to be 1109.47 km^2^, and the proportion of construction land is 30.88%. Comparing natural development with the ecological security scenario, the latter scenario has a lower net increase in the area of construction land and a lower proportion of construction land. In addition, the AI of the latter is 20.22, which is significantly lower than that of 2000–2010 and 2010–2018. The AGR of the latter is also lower (2.08%). Under the ecological security scenario, the expansion area, AI and AGR of construction land are all less than those under the natural development scenario, indicating that the ecological security pattern can effectively constrain urban expansion.

From 2000 to 2030 (Figure 8), under both scenarios, construction land maintains an expansion trend, but the expansion area would be smaller under the ecological security scenario. AI maintains increasing under the natural development scenario, but under the ecological security scenario, AI would drop quickly from 2018 to 2030. AGR maintains declining, and under the ecological security scenario, it would decline faster.

#### 3.3.3. Urban Expansion in Different Directions

Under the natural development scenario, the area of construction land (Figure 9a) in all directions is expanding. By 2030, the net increased area of construction land in the northwest and southeast directions will exceed 90 km^2^. The proportion of construction land in each direction exceeded 33%, and the area of construction land in the southeast direction accounted for 39.94%. Under the ecological security scenario, the construction land in each direction also keeps expanding, but the net increase area in each direction would be less than 80 km^2^, and the value in the southwest direction is expected to be the smallest, which is 31.96 km^2^. In other directions, the construction land would account for more than 30%, of which the construction land in the southeast direction would account for 38.23%.

In the northeast direction, AI and AGR decrease under both scenarios (Figure 9b,c), but under the ecological security scenario, AI and AGR decrease more significantly. In the northwest direction, under the natural development scenario, AI is increasing, while it is decreasing under the ecological security scenario. In this direction, there exists contiguous woodland, which would be converted into construction land under the natural development scenario but would not change under the ecological security scenario because of the ecological security pattern. The AGR is decreasing under both scenarios, and under the ecological security scenario, the value would be lower. In the southwest direction, AI declined under both the natural development scenario and the ecological security scenario. Furthermore, the AI of the latter would decline more significantly. Compared with other directions, AI will be the lowest in this direction; the value of the former would be 5.61, and the latter would only be 2.66. Because of the large number of ecological sources, under the ecological security scenario, construction land expansion in this direction would be more difficult. The change in AGR would be similar to AI, but the AGR under the natural development scenario will not be the lowest in the four directions. In the southeast direction, AI increased under both scenarios, but under the natural development scenario, AI was higher. In addition, under this scenario, AGR would increase, while under the ecological security scenario, the value would not increase significantly.

Compared with the natural development scenario, the AI in each direction would be lower under the ecological security scenario. AI in the southwest direction is the lowest under both scenarios. A large number of ecological sources are distributed in the southwest direction, so under the ecological security scenario, the expansion of construction land in this direction would be hindered. Under the natural development scenario, only in the southeast direction would the AGR be slightly higher than that in the previous stage, and in other directions, it would not be much different; that is, the expansion speed of construction land in each direction would be similar. Under the ecological security scenario, the AGR in each direction would be less than that in the previous stage, and the expansion rate in the southwest direction would be the lowest, which is in accordance with the distribution of the ecological security pattern.

#### 3.3.4. Urban Expansion from City Center

Under the natural development scenario, the area of construction land (Figure 10a) within 20 km would exceed 50%, and construction land would begin to appear in the range of 55–60 km. Under the ecological security scenario, the proportion of construction land within 15 km is expected to exceed 50%, and it would be 48.76% within 15–20 km. In all ranges, the construction land area under the ecological security scenario would be smaller.

Under the natural development scenario, within 20 km, the AI decreases (Figure 10b), while at 20–30 km, it tends to decrease, especially in the range of 25–30 km, indicating that the expansion intensity of the central area of the study area tends to weaken and that the development focus of construction land will gradually move outwards. Under the ecological security scenario, the expansion intensity of construction land in each range shows the same trend, but AI would be less than that under the natural development scenario, indicating that the ecological security pattern can effectively weaken the expansion intensity of construction land. In the range of 30–45 km, the AI would increase slightly under the natural development scenario but would decrease slightly under the ecological security scenario. In other ranges, the AI would not change obviously.

Under the natural development scenario, the AGR of construction land (Figure 10c) within the range of 25 km would slow significantly, while in areas beyond 25 km, construction land would maintain a relatively high growth rate. Especially in the range of 45–50 km, the AGR exceeds 13%, but because of less original construction land, the actually increased construction land area would be very small. It is evident that the construction land is expanding rapidly in the area far from the center. Under the ecological security scenario, the changing trend of AGR is similar to the natural development scenario, but in each range, the AGR would be lower than the former. This shows that the ecological security pattern would restrict the expansion speed of construction land.

## 4. Discussion

Scenario analysis can be used to predict and evaluate regional land use change under different development scenarios [42,43], which could provide important scientific references for formulating regional development policies. With the MSPA method and MCR model [32], this study constructs an ecological security pattern of the downtown of Hangzhou and simulates future urban expansion under the natural development and ecological security scenario. The study of current multi-scenario simulation based on the ecological security pattern pays more attention to its impact on the urban ecology, and few studies pay attention to its impact on direct urban changes. This study compared and analyzed the indexes of construction land expansion in different directions and distances from the city center, including area, proportion, AI, and AGR, which intuitively showed the characteristics of regional construction land expansion under different development scenarios.

Ecological security patterns can effectively restrict the expansion of construction land [27], which is of great significance to the optimization of urban patterns. Current domestic ecological protection policies are mainly based on the ecological protection red line [44], which evaluates the ecological importance mainly from two aspects, which are ecosystem service importance and ecological sensitivity, then identifies important ecological areas, and human activities are strictly regulated corresponding to the importance (*the technical guide for the delimitation of ecological protection red line*). This policy is obviously positive for ecological protection, but it focuses on key patches and ignores the connectivity among them. Although the government of Hangzhou has proposed building ecological corridor nets before, there is no specific policy to protect them. In addition, small ecological patches, which are ecological nodes in this study, should also be considered the “stepping stone” of biological migration to attach more importance. Many scholars in the world have realized that the green infrastructure has a positive effect on the sustainable development of cities [45,46,47], and proposed to incorporate it into urban planning [48,49], such as green belt [50], urban park, etc. However, most people believe that the green infrastructure is local, and few people build an ecological framework from the overall situation to promote regional ecological sustainability. This study based on the whole study area, builds an overall ecological framework for the rapid urbanization area, in order to find a way to balance urban expansion and ecological protection.

In this study, the ecological security pattern was unevenly distributed spatially. Ecological sources were dominant in the ecological security pattern, and woodlands were the main component. Woodland was contiguously distributed in the west and north of the study area, construction land occupied the central area, arable land occupied most of the remaining space, and others were scattered. Important ecological land types spatially corresponded to the ecological security pattern. Representative rivers in the study area, mainly the Qiantang River, were also included in the ecological corridor net to help achieve ecological exchange between the east and west of the study area and to increase the connectivity of ecological security patterns. Overall, under the condition of the extremely unbalanced distribution of ecological resources in the study area, the integrity and connectivity of the ecological security pattern constructed in this study have reached a good level.

Comparing the expansion of construction land under two scenarios and the land use change in the past 20 years, it was found that the construction land in the study area is expected to continue to expand, but its expansion would be slower in recent years, especially under the ecological security scenario. In addition, under the natural development scenario, the expansion of construction land would be more balanced, while under the ecological security scenario, because of the large proportion of ecological sources in the region, the construction land would be obviously limited in the southwest direction, and urban expansion would be unbalanced. In terms of the expansion distance, the focus of the expansion of construction land is moving outwards from the center. With a high proportion (nearly 50%) of construction land, there tends to be less new construction land, and the city can only sprawl outwards, which is in accordance with the current spatial development trend of most cities in China. In practice, due to strict restrictions on newly added construction land and urban expansion, as well as global comprehensive renovation (construction land would be renovated into other land types such as arable land and grassland), it is more likely to renovate existing construction land rather than add new. The proportion, AI, and AGR of construction land might be lower. In view of the disorderly expansion of cities, currently, the most important policy in China is the urban development boundary, which is the largest boundary that allows the expansion of urban construction land and spatially limits the maximum scope of urban expansion. Strong public policies can effectively control urban expansion. For the sustainable development of cities, while restricting the outward urban expansion, it is necessary to promote inward development to make full use of developed land and to avoid encroaching on ecological space.

There are also some deficiencies in this study. In the process of constructing the ecological security pattern, although the ecological prospect selected the land types with high ecological value, the MSPA method paid more attention to the shape of patches, and focuses on the patches with a large area, some small patches with high ecological value may be ignored. Further studies may add ecological value evaluations [51,52] of ecological patches to amend it. Besides, although the MCR model has been widely used to construct ecological corridors, selecting important ecological corridors with the gravity model may ignore some corridors of great significance to small areas. In addition, in the simulation of urban expansion, different parameter settings such as neighborhood may also slightly affect the simulation results. Due to the difficulty of acquiring land use data in 2020, this study took the land use data in 2018 as the base period to conduct a multi-scenario simulation in 2030. Different time spans may lead to some differences in simulation results, but the overall trend is consistent, which can still support that the ecological security pattern can effectively constrain urban expansion and does not affect the research conclusion.

## 5. Conclusions

Ecological security patterns are of great significance to the optimization of urban patterns. In this study, the downtown area of Hangzhou was selected as the study area, and the ecological security pattern was constructed by the MSPA method and MCR model. A large number of ecological resources were distributed in the west of the region, which implied that the ecological resources in the study area were unevenly distributed from east to the west and that the resources in the west were obviously richer. With the help of the natural ecological corridors, the Qiantang River, East–West ecological exchange in the study area can be realized.

From 2000 to 2018, the area of construction land was in a stage of rapid expansion, the intensity of urban expansion was increasing, and the rate of urban expansion slowed down. From the perspective of expansion direction, it was relatively balanced in all directions, but the expansion in the southeast direction was more significant. In terms of the distance from the city center, although the central area had a relatively high proportion of construction land, its AI and AGR tended to decline. By contrast, AI and AGR within a range of 20–30 km increased, indicating that the construction land was likely to move outwards.

Based on the fact that construction land in the study area has expanded rapidly in the past few decades, this study proposed two scenarios, the natural development scenario and the ecological security scenario, to verify the impact of ecological security patterns on urban expansion. In terms of urban expansion direction, under the ecological security scenario, because of the restriction of ecological security patterns, the expansion of construction land in the southwest direction would be difficult. In addition, the AI and AGR under the ecological security scenario tend to be lower than those under the natural development scenario, which means that the newly added construction land area in the future would be smaller. The analysis showed that the ecological security pattern can effectively control urban expansion and has a positive impact on the optimization of urban patterns. Many cities have the problem of urban expansion currently, although this study focused on Hangzhou, it could be a reference for other cities.

Urban expansion is a long-term and widely concerned topic, which has caused a series of problems globally. Due to urban expansion, the environment has deteriorated in Arapiraca City, Brazil [53], the above-ground carbon loss in Zanzibar City, Tanzania [54], and peri-urban farmers’ poverty has increased in Africa [55], etc. Green infrastructure has been used to address urban expansion for many years [56]. The advancement towards urban green infrastructure (UGI) planning is well established and progressing in Europe [57]. The Metropolitan Green Belt (MGB) has successfully restricted the outward physical growth of London [50]. In addition, urban planning is also an effective measure of urban expansion. Policymakers must consider urban growth factors under different land cover scenarios, monitor the impacts of urban expansion on urban ecology [58], develop urban greenery strategy and support urban densification [54], assess urban ecosystem services at the right scale and resolution [59], account for the multi-functionality of urban green infrastructure, define strategic objectives, ensure long-term commitment in the implementation phase, and strengthen planning arguments against conflicting interests [60].

## Figures and Tables

**Figure 1 ijerph-19-00301-f001:**
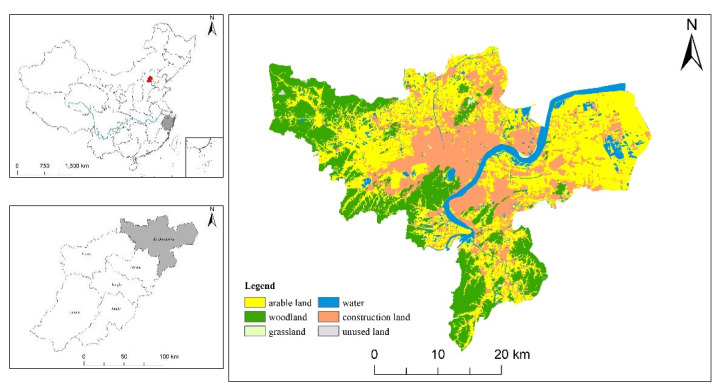
Location of the study area.

**Figure 2 ijerph-19-00301-f002:**
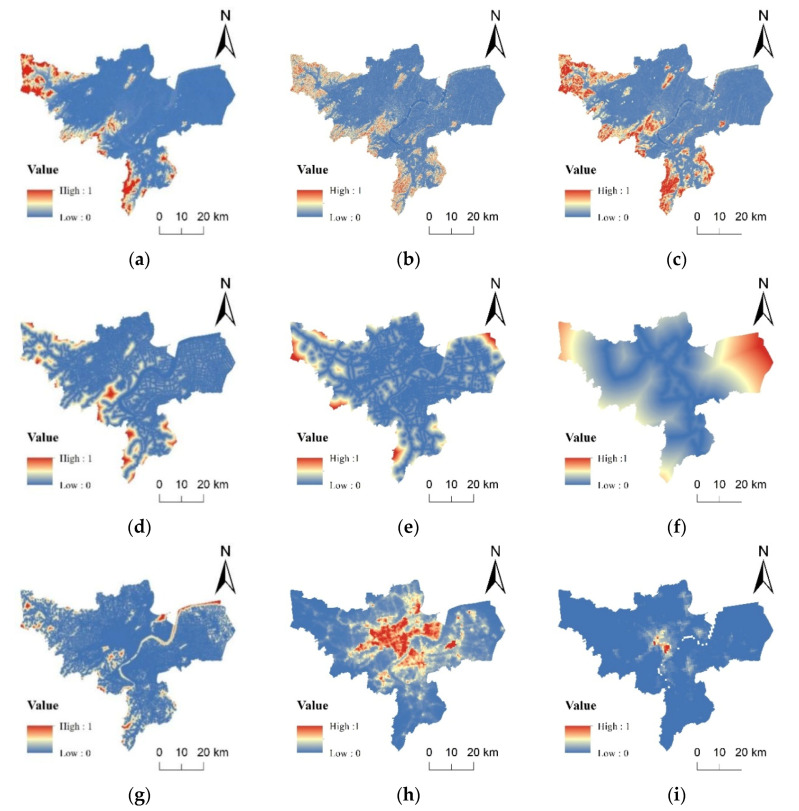
Data of driving factors: (**a**) elevation; (**b**) slope; (**c**) topographic relief; (**d**) distance to rivers; (**e**) distance to roads; (**f**) distance to railways; (**g**) distance to residential area; (**h**) night light data; (**i**) population density.

**Figure 3 ijerph-19-00301-f003:**
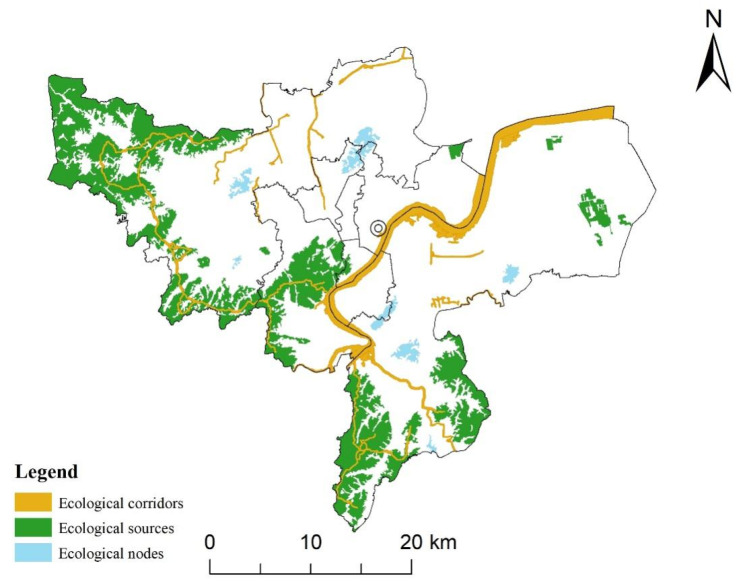
Spatial distribution of ecological security pattern.

**Figure 4 ijerph-19-00301-f004:**
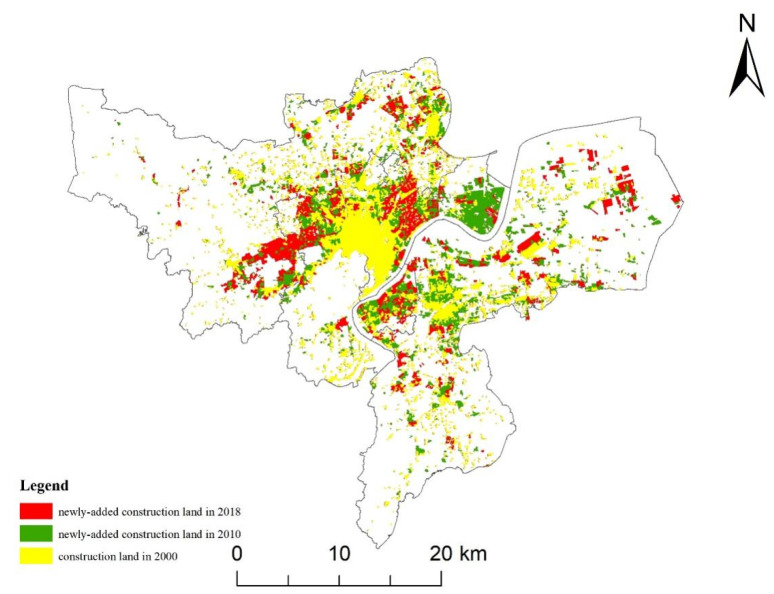
Spatiotemporal changes in construction land in 2000, 2010, and 2018.

**Figure 5 ijerph-19-00301-f005:**
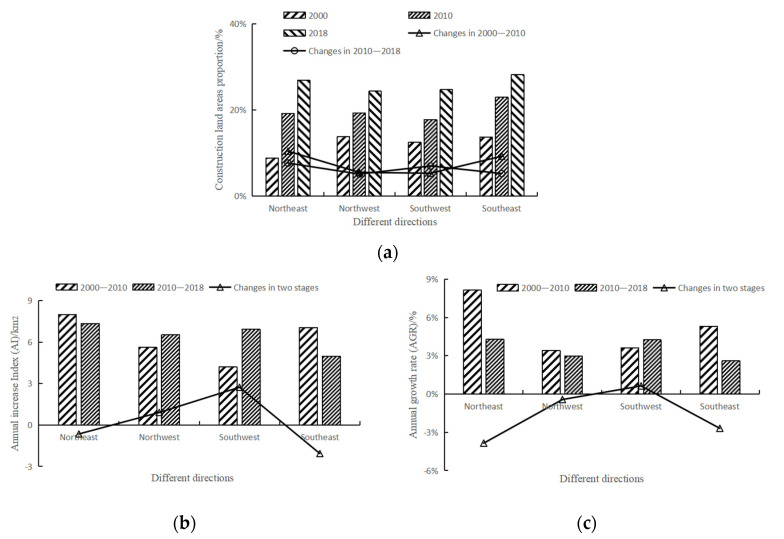
Characteristics of urban expansion in different directions in 2000–2018: (**a**) construction land area proportion; (**b**) annual increase index (AI); (**c**) annual growth rate (AGR).

**Figure 6 ijerph-19-00301-f006:**
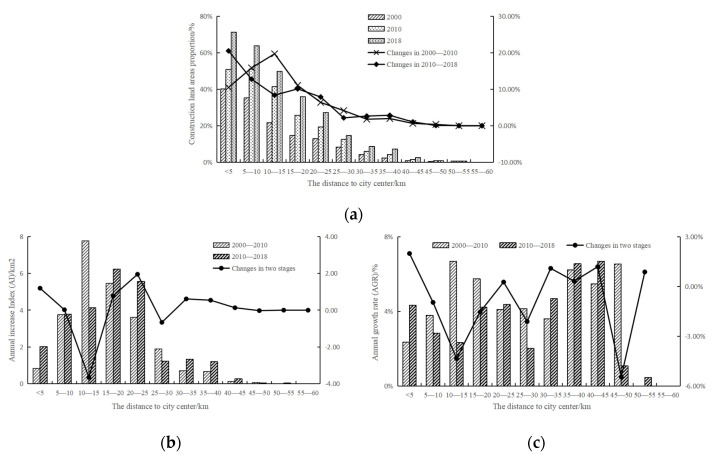
Characteristics of urban expansion from the city center in 2000–2018: (**a**) construction land area proportion; (**b**) annual increase index (AI); (**c**) annual growth rate (AGR).

**Figure 7 ijerph-19-00301-f007:**
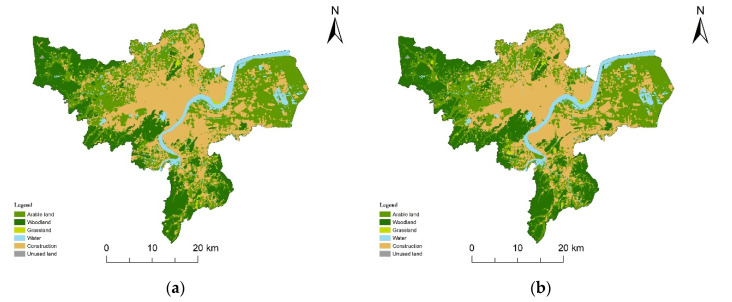
Land use data in 2030 under different scenarios: (**a**) natural development scenario; (**b**) ecological security scenario.

**Figure 8 ijerph-19-00301-f008:**
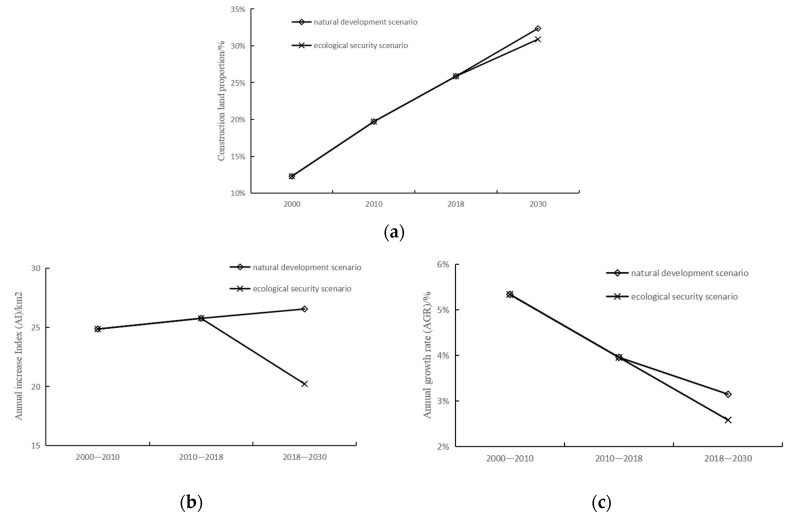
Characteristics of urban expansion from 2000–2030: (**a**) construction land area proportion; (**b**) annual increase index (AI); (**c**) annual growth rate (AGR).

**Figure 9 ijerph-19-00301-f009:**
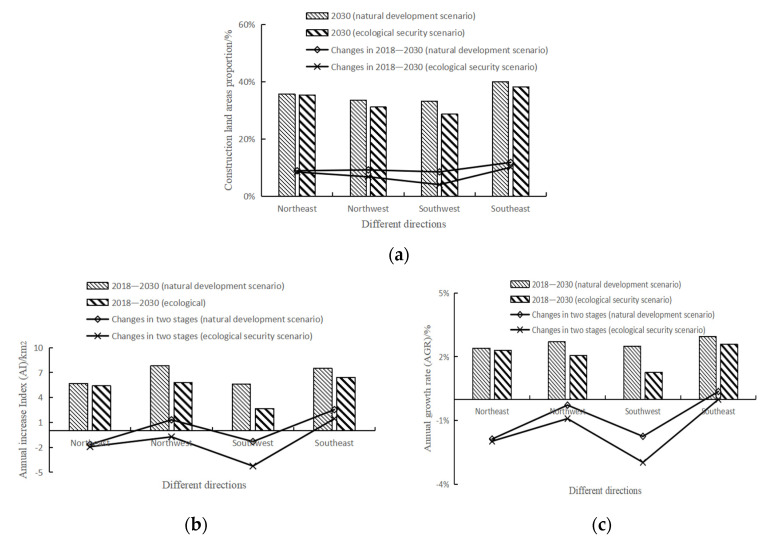
Characteristics of urban expansion in different directions under different scenarios: (**a**) construction land area proportion; (**b**) annual increase index (AI); (**c**) annual growth rate (AGR).

**Figure 10 ijerph-19-00301-f010:**
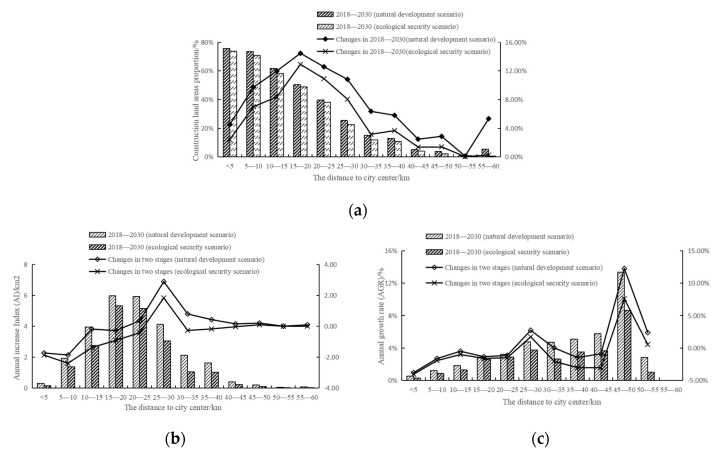
Characteristics of urban expansion in different directions under different scenarios: (**a**) construction land area proportion; (**b**) annual increase index (AI); (**c**) annual growth rate (AGR).

**Table 1 ijerph-19-00301-t001:** Ecological resistance score.

Indicator Types	Classification	Value	Weight
land use/cover types	Woodland	1	0.7
	Water	1	
	Grassland	30	
	Arable land	50	
	Unused land	90	
	Construction land	100	
topographic relief	<8	90	0.3
	8–17	70	
	17–27	50	
	27–37	30	
	37–50	10	
	>50	1	

**Table 2 ijerph-19-00301-t002:** Neighborhood weight.

Arable Land	Woodland	Grassland	Water	Construction Land	Unused Land
0.1	0.23	0.35	0.5	1	0.4

**Table 3 ijerph-19-00301-t003:** (**a**) Land transfer parameters under the natural development scenario. (**b**) Land transfer parameter under the ecological security scenario.

(a)						
Land Types	Arable Land	Woodland	Grassland	Water	Construction Land	Unused Land
Arable land	1	1	1	1	1	0
Woodland	1	1	1	0	1	0
Grassland	1	1	1	0	1	1
Water	1	1	1	1	1	1
Construction land	1	1	1	0	1	0
Unused land	1	1	1	1	1	1
**(b)**						
**Land Types**	**Arable Land**	**Woodland**	**Grassland**	**Water**	**Construction Land**	**Unused Land**
Arable land	1	1	1	1	1	0
Woodland	0	1	1	0	0	0
Grassland	0	1	1	0	1	1
Water	1	1	1	1	1	1
Construction land	1	1	1	0	1	0
Unused land	1	1	1	1	1	1

**Table 4 ijerph-19-00301-t004:** Indicator changes in construction land expansion in 2000, 2010, and 2018.

Urban Expansion Indicators	2000	2010	2018
Construction land areas/km^2^	411.52	660.08	866.18
Construction land proportion/%	12.29	19.71	25.86
Annual increase index (AI)		24.86	25.76
Annual growth rate (AGR)		4.84	3.46

**Table 5 ijerph-19-00301-t005:** Characteristics of urban expansion from the city center in 2000–2018.

Urban Expansion Indicators	2030 (Natural)	2030 (Ecological)
Construction land areas/km^2^	1185.81	1109.47
Construction land proportion/%	32.35	30.88
Annual increase index (AI)	26.55	20.22
Annual growth rate (AGR)	2.64	2.08

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
