# Peer review of "Simulating Urban Expansion Based on Ecological Security Pattern—A Case Study of Hangzhou, China"

_ijerph, 2021, doi:10.3390/ijerph19010301_

Round 1

Reviewer 1 Report

The work carried out by the authors shows great interest in the topic "Ecological Security Pattern". Thanks for the effort made.

It is recommended to improve your writing and language.

In the abstract section, it is recommended to add the objective of the study.

The abstract section describes the results that do not indicate which scenario they belong to.

The manuscript does not adequately present the methods used and the results.

The results section is extensive, losing interest in reading. Some paragraphs do not contribute to the study. In addition, several results (i.e., growth and ecological nodes) are not appreciated in the figures and tables in this section. The quality of the figures and tables should be improved.

Line 94. In several sections of the manuscript the concept of ecologically civilized cities is used. It is recommended to add its meaning.

Line 116. Some elements described in lines 97-115 are not represented on the map. For example: urban areas and current ecological lands.

Line 129. It is important to mention the resampling method used, because the change in spatial resolution from 1000 m to 30 m has effects on the original information.

Line 141. What are the authors referring to by this phrase "has an obvious scale effect"?

Line 144. How were the Delta values ​​obtained? What is its usefulness in calculating probabilities?

Lines 161 to 172. It is recommended to improve the wording. It is difficult to understand the procedure used.

Line 163. What is the gravity model? and How was it used at work?

Lines 206 to 217. The text is closer to the results than the description of the methods.

Line 218. It is recommended to improve the figures.

- Why is map "f" different from other maps? Is the method for its design different? Explain it

Line 234. The elements of the formula are not described.

Author Response

Thank you for your suggestions gratefully. We have revised the manuscript as required, and the attachment is the point by point response. If you have any questions, please do not hesitate to contact us. Thanks again, and wish you have a nice day.

Reviewer 2 Report

The article deals with an interesting research problem. It clearly presents the methods and results. I suggest that the authors consider the following changes:
- I miss an isolated review section (I do not include the Introduction). Maybe it would be worth preparing such a part. This would allow the authors to characterise the topic even better, more generally, from the perspective of the international discussion
- the Discussion also lacks a broader reference to international theses in the literature on the subject. I suggest a strong expansion of this section and an even broader emphasis on the novelty of the research.

Author Response

(The authors gave the same response as above.)

Round 2

Reviewer 1 Report

I thank the authors for the changes made to the manuscript.

This manuscript is a resubmission of an earlier submission. The following is a list of the peer review reports and author responses from that submission.